# Environmental Risk Assessment for rVSVΔG-ZEBOV-GP, a Genetically Modified Live Vaccine for Ebola Virus Disease

**DOI:** 10.3390/vaccines8040779

**Published:** 2020-12-19

**Authors:** Joan G. Tell, Beth-Ann G. Coller, Sheri A. Dubey, Ursula Jenal, William Lapps, Liman Wang, Jayanthi Wolf

**Affiliations:** 1Merck & Co., Inc., Kenilworth, NJ 07033, USA; Beth-ann.coller@merck.com (B.-A.G.C.); sheri_dubey@merck.com (S.A.D.); william.lapps.jr@merck.com (W.L.); liman_wang@merck.com (L.W.); jayanthi_wolf@merck.com (J.W.); 2Jenal & Partners Biosafety Consulting, 4310 Rheinfelden, Switzerland; ursula.jenal@jenalpartners.ch

**Keywords:** ERA, rVSV, recombinant vaccine, GMO, vesicular stomatitis virus, *Zaire ebolavirus*, shedding, viremia, environmental impact, ERVEBO^®^

## Abstract

rVSVΔG-ZEBOV-GP is a live, attenuated, recombinant vesicular stomatitis virus (rVSV)-based vaccine for the prevention of Ebola virus disease caused by *Zaire ebolavirus*. As a replication-competent genetically modified organism, rVSVΔG-ZEBOV-GP underwent various environmental evaluations prior to approval, the most in-depth being the environmental risk assessment (ERA) required by the European Medicines Agency. This ERA, as well as the underlying methodology used to arrive at a sound conclusion about the environmental risks of rVSVΔG-ZEBOV-GP, are described in this review. Clinical data from vaccinated adults demonstrated only infrequent, low-level shedding and transient, low-level viremia, indicating a low person-to-person infection risk. Animal data suggest that it is highly unlikely that vaccinated individuals would infect animals with recombinant virus vaccine or that rVSVΔG-ZEBOV-GP would spread within animal populations. Preclinical studies in various hematophagous insect vectors showed that these species were unable to transmit rVSVΔG-ZEBOV-GP. Pathogenicity risk in humans and animals was found to be low, based on clinical and preclinical data. The overall risk for non-vaccinated individuals and the environment is thus negligible and can be minimized further through defined mitigation strategies. This ERA and the experience gained are relevant to developing other rVSV-based vaccines, including candidates under investigation for prevention of COVID-19.

## 1. Introduction

Ebola virus disease (EVD), caused by Ebola virus (EBOV; species: *Zaire ebolavirus*), carries an extremely high case fatality rate, ranging from 40 to 70% in several outbreaks over the last decade [1]. EVD survivors frequently suffer from debilitating long-term sequelae, including musculoskeletal pain, neurocognitive deficits, depression, fatigue, ocular disorders, and immune dysfunction [2,3,4,5]. Apart from isolated laboratory accidents, all outbreaks originated from rural regions of Western and Middle Africa [1]. Previous EVD outbreaks, in particular the 2014–2016 epidemic, led to devastating consequences for affected communities, regional/national economies, and health care systems [1,6]. Of grave concern is the continuing potential for local outbreaks to progress to global spread.

EVD is a zoonotic disease, with EBOV believed to be transmitted to human index cases through close contact with bodily fluids of infected, rainforest-dwelling mammals. Prevailing evidence suggests that fruit bats are the predominant natural host reservoir of EBOV, with other species such as rodents, porcupines, pigs, dogs, non-human primates, and also humans merely serving as accidental spillover hosts (Figure 1a) [1,7,8,9,10]. However, once the virus has successfully infected an index patient, EBOV readily spreads human to human by direct contact with bodily fluids/tissues of EVD patients or with fomites contaminated with such infectious materials [1]. EBOV entry into host cells is mediated by EBOV glycoprotein GP, the virus’ only envelope protein, which binds to the Niemann–Pick C1 receptor involved in intracellular cholesterol transport [11,12,13,14]. While the virus can infect most cell types, its primary targets are dendritic cells and mononuclear phagocytes. EVD progresses rapidly: further recruitment of EBOV-susceptible immune cells to infected tissues leads to immune system dysregulation, extensive lymphocyte destruction, and endothelial barrier breakdown, eventually resulting in multiple organ dysfunction and death [1]. Due to the very high mortality, substantial long-term morbidity and sequelae (e.g., cataracts in children), destructive community-level consequences, and epidemic potential associated with EBOV, EVD vaccines that can be readily employed to combat future outbreaks are critical [6].

The first EVD vaccine to be approved and widely used in African countries was rVSVΔG-ZEBOV-GP, a live, attenuated, recombinant vesicular stomatitis virus (rVSV)-based vaccine specific to EBOV [33]. In this chimeric virus vaccine, the wild-type gene sequence for the VSV envelope glycoprotein (VSV-G) has been deleted completely and replaced with the gene sequence encoding EBOV GP (Figure 2). No other sequence from EBOV (or any other source) is inserted. Approval of rVSVΔG-ZEBOV-GP for prevention of EVD by the United States Food and Drug Administration (FDA), the European Medicines Agency (EMA), and several African countries’ regulatory bodies (as well as WHO prequalification) occurred just 5 years after initiation of the first phase 1 trial. This accelerated timeline, made possible in large part through unprecedented public–private collaboration between multiple, diverse stakeholders around the world [34], was faster than is typical for vaccine development [35]. Several phase 2/3 clinical trials successfully demonstrated the efficacy, safety, and good tolerability of rVSVΔG-ZEBOV-GP [36,37,38,39], including the cluster-randomized, ring vaccination trial “Ebola ça Suffit!” conducted in the real-world setting of the 2013–2016 West African Ebola epidemic [40]. The vaccine is administered as a single dose [33]. With this single-dose administration schedule, rVSVΔG-ZEBOV-GP is highly immunogenic: more than 90% of recipients generate IgG-binding antibodies, with peak levels (i.e., geometric mean titers of ~1000) at 4 weeks post-vaccination. Antibody titers remain elevated for at least 2 years and immune responses to the vaccine are consistent across age groups [41,42,43,44,45]. The vaccine has a positive benefit–risk profile, reflected in the low rate of serious vaccine-related adverse events and no vaccine-related deaths [36,37,39,43,44,46,47,48]. Since the frequency and magnitude of future EVD outbreaks cannot be predicted, global public health preparedness to quickly combat the disease when it arises is crucial, and rVSVΔG-ZEBOV-GP is an important medical countermeasure and component of readiness strategies.

The VSV that serves as the backbone for rVSVΔG-ZEBOV-GP is a single-stranded RNA virus from the *Rhabdoviridae* family. The ability of recombinant VSV to induce strong, protective humoral and cellular immune responses after a single dose has led to considerable interest in developing rVSV as a vaccine vector [49,50]. Wild-type VSV (wtVSV) VSV can infect humans, in whom it is either asymptomatic or causes mild, acute, influenza-like illness lasting 3–6 days (often with no vesicle formation), without any known complications or deaths [15,41,51]. However, VSV causes clinically significant disease in cattle, horses, and pigs and can result in substantial economic losses to livestock producers [15,51,52,53,54,55]. Besides domestic livestock, VSV is able to infect an extremely wide range of hosts, including many species of vertebrates and arthropods. VSV is endemic to tropical and subtropical regions of Central and South America, but also infrequently (in approximately 10-year cycles) causes epizootics in the United States. It is currently not considered to be naturally present outside the Americas [15,51,53,56,57]. Disease in livestock is self-limited with low mortality and manifests as crusting and vesiculation of mucous membranes and skin, predominantly the tongue, gums, lips, teats, and coronary bands of the hooves. These properties make VSV a naturally attenuated vector backbone upon which human vaccines can be developed [41]. The two main serotypes of VSV are ‘New Jersey’ (NJ) and ‘Indiana’ (I); rVSVΔG-ZEBOV-GP is based on VSV-I, which is less virulent than VSV-NJ [53,58,59]. Cell entry of wild-type VSV (wtVSV) is mediated by VSV-G, a type III viral membrane fusion protein that binds to the ubiquitous low-density lipoprotein [1] receptor [60,61,62]. Since members of the LDL receptor family can be found throughout the animal kingdom in virtually all cell types, VSV exhibits broad host and cell tropism. Mammalian cells particularly susceptible to VSV infection include keratinocytes, monocytes, and other MHC-II-positive cells, as well as various cell types in the central nervous system and respiratory tract [41]. The complete substitution of VSV-G by EBOV-GP significantly impacts both the host and cellular tropism of the recombinant vaccine compared with wtVSV-I. For example, rVSVΔG-ZEBOV-GP shows none of the neurovirulence associated with wtVSV [34,41,63]. The overall virulence of rVSVΔG-ZEBOV-GP is also reduced compared with wild-type virus [41,50,64,65,66,67]. The main underlying factor behind this attenuation is the deletion of the VSV-G protein, a key determinant of VSV pathogenicity.

VSV epidemiology and ecology, including the virus’ natural host reservoir species (i.e., a vertebrate species capable of developing sufficiently high viremia to infect hematophagous vectors) and its transmission cycle, are only incompletely understood [15,68]. Vertebrates, which exhibit significant viral shedding through skin and mucous membrane lesions and are also potentially capable of sustained VSV viremia, are thought to act as amplifying reservoirs [15]. Arthropods in turn facilitate viral transmission by acting as either biological or mechanical vectors [15]. Among wild mammals, VSV antibodies have been detected in species as diverse as monkeys, rodents, deer, and bats [16,17,18,20,53]. Within a mammalian herd or group, VSV spread occurs efficiently from one infected animal to another via direct contact, aerosols, or fomites. The key role of various blood-feeding insects as transmission-competent vectors in the VSV life cycle was confirmed by a number of epidemiologic and laboratory studies. These biological vectors predominantly include so-called ‘pool feeders’, i.e., phlebotomine sandflies (*Lutzomyia* and *Phlebotomus* spp.), black flies (*Simulium* spp.), and biting midges (*Culicoides* spp.); however, mosquitoes (which are ‘capillary vessel feeders’) may also be involved [15,19,22,23,24,26,27,28,68]. Since there is generally only low, transient, or no VSV viremia in most infected vertebrates (other than experimentally infected rodents) [29,30], infection of biological vectors via ingestion of blood from a VSV-positive host is unlikely to be an important transmission route, albeit still possible [15]. Instead, infection of hematophagous vectors may occur either via co-feeding with already infected insects or via feeding around or in vesicular skin lesions [15,31]. Unlike mosquitoes, which penetrate directly into capillary blood vessels using their proboscis, pool feeders’ mouthparts cut into their host’s skin and then ingest the pooled blood. This makes the latter considerably more likely to come into contact with contaminated epidermal surfaces [15]. Migratory grasshoppers are also believed to serve as an VSV reservoir and as long-distance mechanical vectors [15,25,69]. In addition, biting and non-biting flies can mechanically transmit VSV through contact with vesicular lesions on an infected animal (e.g., by feeding on secretions) and subsequent transfer of infectious material to a healthy host [15]. Experimental evidence suggests that mosquitoes and flies can vertically transmit VSV to their embryonated eggs [70] and that biting midges can transmit the virus venereally [32].

Given these multiple modes of transmission, the wide range of susceptible host species, and the different types of insect vectors, the ecology of VSV is exceedingly complex. One working hypothesis of its natural life cycle in tropical regions (Figure 1b) involves sandflies as the biological vector, with the virus being vertically transmitted from adults to their offspring. Sandflies are then thought to spread the virus to small rodents (e.g., deer mice, cotton rats) and other small mammals, in which the virus is amplified and that act as the main natural reservoir. Viremic small mammalian hosts then serve as an infection source for the sandfly vectors. In addition, a variety of other insect species may act as additional mechanical and/or biological vectors to transmit the virus to livestock and humans [15,16,17,19,21,29,30,31,41].

## 2. Environmental Assessment Requirements for GMO Vaccines

Since rVSVΔG-ZEBOV-GP is a live (i.e., attenuated, but replication competent), genetically modified organism [2], environmental assessments were required to secure approvals for clinical trial and marketing authorizations. This is because live GMO vaccines could have adverse impacts on humans, animals, plants, microorganisms, food webs, and/or ecological processes if they were to inadvertently enter the natural environment. Such harmful impacts may be direct or indirect, immediate or delayed. The worst-case scenario would be if a GMO medicinal product entered the environment, became widespread, and underwent genetic changes that increase its potential to cause various harms to human or environmental health.

As part of license applications for recombinant vaccines, the FDA requires either an environmental assessment or, alternatively, a justification why a full, extended environmental assessment is not necessary—a so-called ‘claim of categorical exclusion’ [71]. Such a claim should demonstrate that the vaccine will not meet any of the following four criteria:Potential effects on the quality of the environment are likely to be highly controversial;Potential effects on human health are highly uncertain or involve unique or unknown risks;May have potential effects on an endangered/threatened species or its habitat;Potential effects may violate federal, state, or local laws or requirements imposed for the protection of the environment.

Similarly, in Japan a biological diversity risk assessment report is required prior to receiving regulatory approval for initiating clinical trials of GMO vaccines. This report must include ecological and physiological information on the wild-type recipient organism, details on how the GMO vaccine is constructed, information on how the vaccine will be used in practice, and a list of proposed mitigation measures to help avoid adverse impacts on biodiversity [72].

Out of the various regulatory requirements pertaining to environmental assessments for rVSVΔG-ZEBOV-GP, the most in-depth and complex was the environmental risk assessment (ERA) requested by the EMA [73,74]. The EMA necessitates ERAs for all drug marketing authorization applications, but ERAs for medicinal products containing a GMO (including live recombinant virus vaccines) have substantially different requirements than those for other treatments, such as small molecules (Table 1) [74]. This is because clinical use of a live recombinant vaccine is considered deliberate release of a GMO into the environment, where it may become irreversibly and uncontrollably established. An ERA must first evaluate the potential environmental harms of the product, subsequently define risk mitigation measures if any unacceptable risks are identified, and finally re-evaluate the environmental risk remaining after implementing these measures. While vaccine recipients’ safety is not the subject of environmental assessments, ERAs must consider harms to other, non-vaccinated individuals. The assessments should also address the various ways in which a live recombinant vaccine could inadvertently enter the environment, including excretion by vaccine recipients, inappropriate disposal of vaccine waste, and/or accidental release during handling or administration of the vaccine [74].

In line with these requirements, environmental assessments for rVSVΔG-ZEBOV-GP thoroughly evaluated potential adverse effects and negative consequences for the environment, as well as for people who may come into contact with immunized persons. Environmental effects were considered mainly in the context of known EBOV and/or VSV host species, but also in terms of potential impacts on other species and environmental processes. Here we summarize results and interpretation of the various preclinical and clinical datasets used to support the EMA ERA for rVSVΔG-ZEBOV-GP and discuss their applicability to potential future vaccines based on the rVSVΔG vector platform, including a COVID-19 vaccine currently in development.

## 3. Developing an ERA for rVSVΔG-ZEBOV-GP

### 3.1. ERA Overview

It is recommended that an ERA compare the recombinant virus vaccine to the naturally occurring virus it is based on, that it be data driven, and be re-evaluated in light of new data that may become available post-completion. A separate ERA should be conducted for each individual vaccine, even for vaccines based on the same platform. However, information learned from similar recombinant vaccines used under similar circumstances can be used to supplement an ERA [74]. An ERA should be conducted in the following step-wise fashion (Figure 3) [73,74]:

Identify any adverse environmental effects that the GMO may cause. In the case of a recombinant virus vaccine, this step should consider the vaccines’ host range, cell and tissue tropism (especially if genes involved in cellular entry are altered, such as with rVSVΔG-ZEBOV-GP), transmission route, infectivity, pathogenicity, replication mechanism, genetic stability, ability to transfer genetic materials to other organisms, and survivability. For this step, it makes most sense to first assess the relevant characteristics of the wild-type virus vector, and then consider and experimentally evaluate how those might be altered by the foreign genetic material [74].Evaluate the potential consequences of each of the identified adverse environmental effects, should it occur. For each of the adverse effects identified in the previous step, the extent of its negative impacts should be classified as high (i.e., significant changes that might affect ecosystem function), moderate, low, or negligible (i.e., no significant changes) on each potentially affected species, ecosystem, or the overall environment [74].Evaluate the likelihood of each identified adverse environmental effect to occur. The likelihood of most negative environmental impacts is difficult, even impossible, to quantify. The classification of likelihoods as high, moderate, low, or negligible is thus useful again. Alternatively, in the previous step, a worst-case scenario could be considered—if the consequences of that scenario can very conservatively be considered as acceptable, then actual quantification in step 3 may not be necessary. For recombinant virus vaccines, preclinical and clinical shedding studies of sufficiently long duration (especially in the case of replication-competent vaccines, such as rVSVΔG-ZEBOV-GP) should be conducted. Vaccine tropism should also be evaluated, which is typically done through in vitro methods assessing the vaccine’s ability to infect different cell types and/or a biodistribution study that quantifies vaccine virus in tissue samples by PCR methods; infectivity assays, such as plaque or TCID50 assays, are then employed to determine if observed virus is replicative [74].Estimate the risk posed by each adverse environmental effect. This step collates the information on magnitude and likelihood of each potential environmental hazard, perhaps in a risk matrix [74]. It is recommended that the risk be regarded as high if there is uncertainty around its likelihood or its consequences [74].Define mitigation strategies to minimize all of the risks associated with the GMO. Mitigation strategies must be defined, at the very least, for all environmental risks deemed as unacceptable in the previous step. Most mitigation measures will seek to minimize the likelihood of negative environmental effects. In the case of live recombinant vaccines, strategies may include adequate disinfection and disposal of materials used in the vaccine-to-recipient supply chain or during actual vaccine administration, protecting the injection site from contact with the environment, and even collection and disposal of the vaccine recipients’ bodily waste. A plan to monitor the effectiveness of mitigation strategies and help identify further, unanticipated hazards should also be developed [74].Determine the GMO’s overall environmental risk. The final step seeks to assess the overall risk by considering the totality of risks posed by each individual environmental hazard in light of the proposed risk management strategies and whether this overall risk profile is acceptable or not [74].

### 3.2. Environmental Risk Potential

#### 3.2.1. Overview of Potential Environmental Issues with rVSVΔG-ZEBOV-GP

As discussed above, the first step in the ERA process was to identify adverse environmental effects that the vaccine may cause. Any potential impact would first require some sort of release of rVSVΔG-ZEBOV-GP into the environment. Release at high titers was deemed unlikely under the recommended transport and administration conditions [33]. However, it is hypothetically possible that vaccinated individuals could transmit the recombinant virus to other humans, domestic livestock, or wild animals. Shed viral particles could also enter the overall environment. Finally, administration accidents could lead to the vaccine infecting health care workers.

According to the relevant EMA regulations, the ERA had to identify potential adverse environmental impacts in the following categories:Could rVSVΔG-ZEBOV-GP cause disease in humans, animals, and/or plants, including allergic or toxic effects?Could rVSVΔG-ZEBOV-GP have effects on the dynamics of populations of species in the receiving environment and the genetic diversity of these populations?Could rVSVΔG-ZEBOV-GP alter susceptibility of any organism to pathogens, thus facilitating the dissemination of infectious diseases and/or creating new reservoirs or vectors?Could rVSVΔG-ZEBOV-GP compromise any prophylactic or therapeutic medical, veterinary, or plant protection treatments?Could rVSVΔG-ZEBOV-GP have effects on biogeochemical cycles, particularly carbon and nitrogen recycling through changes in soil decomposition of organic material?

Each of these potential impacts is discussed in detail below, along with an evaluation of their potential consequences (i.e., step 2 of the ERA) and likelihood of occurrence (i.e., step 3).

#### 3.2.2. Potential to Cause Disease

Preclinical and clinical studies conducted to evaluate virulence and pathogenicity of rVSVΔG-ZEBOV-GP formed the bulk of the data assembled for the ERA. In general, viral transmission requires sufficiently high levels of viremia or viral shedding. Transmission of rVSVΔG-ZEBOV-GP may occur through a susceptible host coming into contact with infected blood, other bodily fluids, or contaminated fomites, hematophagous arthropods acting as biological vectors, or infectious excretions that enter the environment. Both viremia and viral shedding were evaluated in phase 1 clinical trials of rVSVΔG-ZEBOV-GP; only trials completed at the time the ERA was developed were included (Table 2). In all trials, real-time reverse transcription PCR was used to detect and quantify viremia and viral shedding (with viral RNA quantified as genome copies/mL).

Vaccine viremia was frequently observed (in up to 100% of participants) in all of these clinical trials. However, shedding of rVSVΔG-ZEBOV-GP in saliva or urine was infrequent in adults across all of these studies. Vaccine shedding, generally at low levels of <1 × 10^3^ genome copies/mL, was rarely detected in adults (<10%) but was seen in approximately 80% of adolescents during the first 7 days after inoculation (Figure 4). Recombinant virus was only rarely recovered from vesicular skin lesions that developed post-vaccination [41,44,47,76]. The greatest degree of viral vaccine shedding was observed in adolescents and children enrolled at a single site in one of the phase 1 trials (i.e., V920-007), with a maximum value of 7 × 10^4^ genome copies/mL detected in saliva of adolescents [77]; of note, this study did not use the same assays employed in other rVSVΔG-ZEBOV-GP clinical trials. Even this maximum observed value is likely not high enough to permit contact transmission [41,47,75]. Based on these data, the risk of spread via shedding is minimal, and person-to-person transmission of rVSVΔG-ZEBOV-GP was not documented in any of the clinical trials.

The highest viremia levels in all trials were observed in the V920-004 study (Figure 5): the maximum value detected was 2.9 × 10^4^ genome copies/mL, which can still be considered as low with a single infectious particle corresponding to approximately 100 genome copies/mL [47]. As was seen in multiple trials, viremia was dose dependent (i.e., more likely to occur with higher vaccine doses) and transient (i.e., only a single case detected at 2 weeks post-vaccination, none thereafter, with a median duration of 2 days). These data suggest that rVSVΔG-ZEBOV-GP results in low-level viremia generally lasting only a few days.

Since VSV (the backbone of rVSVΔG-ZEBOV-GP) is a vector-borne virus, it is hypothetically possible that a hematophagous insect could transmit the vaccine from an immunized person with sufficiently high vaccine viremia to another human or even into the environment [68]. Of note, viremia in vaccinated humans normally remains below levels (<1000 pfu/mL) required for such biological vector transmission [41]. In order to explore the likelihood of such transmission occurring, the ability of rVSVΔG-ZEBOV-GP to replicate in known hematophagous insect vectors of VSV was assessed in several experiments. The first experiment compared the replication kinetics of rVSVΔG-ZEBOV-GP to those of wtVSV-I in cell lines from *Aedes albopictus* and *Anopheles gambiae* mosquitoes, *Culicoides sonorensis* biting midges, and *Lutzomyia longipalpis* sand flies. While wtVSV-I was able to replicate in each of these in vitro cell cultures, no replication of rVSVΔG-ZEBOV-GP was observed [68].

The second set of experiments evaluated whether rVSVΔG-ZEBOV-GP could infect and disseminate from live *Aedes aegypti* and *Culex quinquefasciatus* mosquitoes exposed to the vaccine through a blood meal or through intrathoracic inoculation. In the intrathoracic inoculation experiments, after 1–2 days wtVSV-I was found in 100% of test animals at mean titers approximately 2–3 times higher than the inoculum dose. On the other hand, rVSVΔG-ZEBOV-GP genomes were detected in approximately one-third of live mosquitoes of both species at titers not exceeding the inoculum. The differences between wtVSV-I and rVSVΔG-ZEBOV-GP levels at that time point were also statistically significant (*p*-values < 0.005). The authors of that study hypothesized that the rVSVΔG-ZEBOV-GP genomes detected were remnants of the original inoculation and not due to replicating virus [68]. This conclusion was supported by additional in vitro data showing that rVSVΔG-ZEBOV-GP genomes were stable for 7 days in *Cx. quinquefasciatus* and for 14 days in *Ae. aegypti* homogenate. Finally, experiments that simulated the way mosquitoes could potentially be infected in a real-world scenario through a blood meal, 2 of 48 *Cx. quinquefasciatus* and 10 of 48 *Ae. aegypti* mosquitoes became infected with wtVSV-I, while none developed an rVSVΔG-ZEBOV-GP infection [68]. The totality of these in vitro and in vivo studies, especially in light of extensive clinical data showing only low-level viremia in vaccinated individuals, therefore strongly suggests two important points: (1) that rVSVΔG-ZEBOV-GP is incapable of significant replication in potential biological vectors, and (2) there is an extremely low probability that hematophagous insects could become infected after feeding on a rVSVΔG-ZEBOV-GP recipient [68]. The authors concluded that there is an extremely low risk of arthropod-borne transmission of rVSVΔG-ZEBOV-GP because it does not possess an arbovirus surface glycoprotein, did not replicate in cell cultures from three different hematophagous insects, and was unable to infect mosquitoes [68].

Besides becoming infected via insect vectors, livestock and wild animals could hypothetically also become exposed to rVSVΔG-ZEBOV-GP through shedding of recombinant virus from vaccinated humans. Among domesticated species, pigs are hypothetically at highest risk of spillover from the recombinant vaccine, because they are susceptible to both wtVSV and wtEBOV. One study conducted in pre-adolescent piglets demonstrated that high-dose intradermal inoculation in the snout did not cause disease and resulted in only minimal shedding of recombinant virus [78]. In a second study conducted in pigs (inoculated both intradermally and oronasally, at high total doses of 4 × 10^7^ pfu), rVSVΔG-ZEBOV-GP caused limited clinical signs consistent with wtVSV infection in some of the test animals, but without any evidence of transmission of vaccine virus from directly infected to contact control animals that were housed together (unpublished data). These results suggest that if a pig were to become infected with rVSVΔG-ZEBOV-GP, there is limited pathogenicity and the risk of further spread to additional animals is very low.

Rodents are also natural hosts of wtVSV and can be experimentally infected with EBOV after the latter virus has been adapted to a rodent host through serial passage. rVSVΔG-ZEBOV-GP, however, was not pathogenic and showed no toxicities in mice and hamsters when given via systemic or intramuscular injection as part of preclinical efficacy (not strictly toxicity) studies [79,80]. This was the case even in severely immunodeficient mice injected with a dose 10 times higher than that used to vaccinate humans. In healthy mice, vaccine given by systemic and mucosal routes at standard doses remained undetectable in blood and organ tissues from 1 to 28 days post-vaccination [79]. Repeat-dose toxicology studies of rVSVΔG-ZEBOV-GP in healthy mice did not show any systemic toxicity, and the vaccine did not produce any developmental or reproductive toxicity in rats (unpublished data).

Similar repeat-dose toxicity studies in cynomolgus monkeys showed histomorphologic findings limited to minimal or mild local injection site inflammation and hyperplasia of draining lymph nodes, each considered to be within acceptable limits and an expected response to intramuscular vaccination (unpublished data). There was also no evidence of neurovirulence following rVSVΔG-ZEBOV-GP intrathalamic brain inoculation in this macaque species [81]. A biodistribution study also done in cynomolgus monkeys showed persistence of vaccine viral RNA in lymphoid tissues, but no infectious virus. Viral RNA after day 7 was generally confined to tissues lacking potential for viral particle shedding, without any evidence of distribution to the central nervous system [41]. 

Finally, in terms of potential vaccine pathogenicity in humans, it is first useful to consider the vaccinated individuals themselves. In clinical trials, the most common systemic adverse reactions were headache, pyrexia, myalgia, fatigue, arthralgia, nausea, chills, arthritis, rash, hyperhidrosis, and abdominal pain. These are the safety issues that non-vaccinated individuals would also be at risk for, should they become infected with the recombinant vaccine. Medical personnel are also at risk of exposure to rVSVΔG-ZEBOV-GP through needle stick injury and could thus experience injection site adverse events observed in clinical trials (i.e., injection site pain, swelling, and erythema); otherwise, the amount of rVSVΔG-ZEBOV-GP from such an injury is much lower than the actual vaccination dose and therefore not expected to bring negative consequences. Any adverse events were generally reported within 7 days after vaccination, were mild to moderate in intensity, and of short duration (less than 1 week). Arthritis was generally reported within the first few weeks post-inoculation, was mostly mild to moderate, and generally resolved within several weeks after onset. It is extremely unlikely that non-vaccinated people potentially infected with the recombinant virus would experience different, more severe, and/or more prolonged adverse events than those seen in trial participants. Overall, the rVSVΔG-ZEBOV-GP vaccine has a favorable benefit–risk profile in humans.

#### 3.2.3. Potential to Affect Populations and Genetic Diversity

The principal way in which rVSVΔG-ZEBOV-GP could impact genetic diversity would be if the vaccine were capable of causing widespread disease in one or more species, resulting in population level decline. The virulence and pathogenicity data discussed above are therefore also pertinent to this aspect of the ERA. Pigs and boars are hypothetically particularly susceptible to rVSVΔG-ZEBOV-GP, since both wtVSV and wtEBOV cause clinical disease in these species. However, even in (domestic) pigs, there was limited evidence of pathogenicity associated with the vaccine, as discussed above. The same lack of pathogenicity was seen in rodents and non-human primates. Finally, arthropods do not seem to become infected with the recombinant virus vaccine and are unable to transmit it as biological vectors. The available data from species representing various orders of mammals and invertebrates therefore do not suggest that rVSVΔG-ZEBOV-GP could have impacts at the population level and/or on genetic diversity.

Genetic diversity may hypothetically also be impacted if the vaccine virus were able to make any genomic changes in host species. However, it is likely impossible that rVSVΔG-ZEBOV-GP could alter, or integrate itself, into eukaryotic or prokaryotic DNA. First of all, the viral RNA genome is closely associated with the virus nucleocapsid protein and replicates outside of the eukaryotic nucleus. Most importantly, however, since there is no DNA phase in its replication cycle and the virus does not encode a reverse transcriptase, there is no possibility that any part of the viral genome could become integrated into a host’s genome or that gene transfer from rVSVΔG-ZEBOV-GP to other species could occur [41,82,83].

#### 3.2.4. Other Potential Effects

It is extremely unlikely that rVSVΔG-ZEBOV-GP would have any other potential effects that the ERA process is intended to evaluate [74]. Since this GMO is a vaccine and had no significant pathogenic effects, it would not facilitate the dissemination of infectious diseases—actually, as a vaccine, it should yield the opposite result. Similarly, rVSVΔG-ZEBOV-GP is not expected to result in the creation of new diseases, disease reservoirs, or vectors, for a number of reasons. First, the recombinant virus has no selective advantage in replication, virulence, or pathogenicity. Second, genetic stability studies indicate that the vaccine’s genome sequence is stable during in vitro replication (i.e., during production of virus seeds and commercial vaccine) [84], suggesting a low probability of point mutations occurring. However, this finding is likely not universally applicable to other rVSV-based vaccines [85], and the genetic stability of each such vaccine virus construct must therefore be evaluated separately. In any case, simple point mutations cannot reverse the attenuation of rVSVΔG-ZEBOV-GP. On the same note, because there are no specific modifications to either the G protein or the VSV backbone that are responsible for attenuation, single mutations cannot cause a reversion of the vaccine to wtVSV or wtEBOV. As single-segment, negative-sense RNA viruses, neither wtVSV nor rVSVΔG-ZEBOV-GP are able to reassert [86,87]. If transmitted, the vaccine virus would therefore retain its attenuated phenotype and likely remain genetically stable. Third, homologous recombination between rVSVΔG-ZEBOV-GP and wtVSV is theoretically possible (i.e., through template switching) in cases of co-infection [86,87,88]. However, the risk of co-infection is very low, because wtVSV is only endemic in some parts of the world, namely Central America and parts of North and South America, and because rVSVΔG-ZEBOV-GP viremia is infrequent and transient. Finally, the probability of non-homologous recombination (with unrelated viruses) is even lower than homologous recombination between related viruses [86,87]. Thus, the generation of new chimeric viruses affecting new animal species is an extremely low-probability, theoretical possibility [89]. If this possibility did manifest, the chimeric construct should not be any more virulent than wild-type virus [90].

### 3.3. Exposure Risk Estimation and Risk Mitigation Strategies

Based on what was previously known about rVSVΔG-ZEBOV-GP, wtEBOV, and wtVSV, as well as considering the experimental data summarized in this report, the risk of any potential adverse effect from the use of rVSVΔG-ZEBOV-GP to non-vaccinated individuals, animals, and the overall environment can be considered negligible. Both the magnitude and the likelihood of exposure are too low to result in adverse effects to rodents, livestock animals, non-human primates, and humans. Of note, the risk to immunocompromised individuals is unknown due to lack of data in this setting.

To minimize this and other unknown/unanticipated risks, several safe management strategies were recommended as part of the ERA (Table 3). Key mitigation measures include personal protective equipment for medical personnel involved in vaccine administration consistent with universal precautions, covering the injection site and any vesicular rash with bandages and/or gauze, and having vaccinated individuals avoid contact with immunocompromised people and livestock for 6 weeks. Successful implementation of many of these strategies depends on good communication between medical personnel administering the vaccine and the vaccinated individuals. If the mitigation measures are not fully understood by the latter, the likelihood of viral particle shedding into the environment is higher. 

### 3.4. Considering Environmental Risks vs. Health Benefits

Based on an evaluation of the magnitude of potential adverse effects and their likelihood of occurrence, the overall risk of rVSVΔG-ZEBOV-GP to human health and the environment is considered negligible. By using appropriate risk management strategies and ensuring that the vaccine is administered under the recommended controlled conditions, exposure of non-vaccinated people and animals to the recombinant vaccine can be prevented altogether. Even if there was transmission from a vaccinated individual, the data suggest minimal or no untoward effects in the recipient organisms and no further spread. Based on the totality of the data, it is highly improbable that non-vaccinated individuals or animals will experience any issues linked to the use of rVSVΔG-ZEBOV-GP.

Regulatory agencies around the world, including the US FDA and the EMA, agreed with these ERA conclusions. The FDA accepted the claim of categorical exclusion and agreed that a full environmental assessment (i.e., identification of environmental effects, assessing their magnitude and likelihood, and recommendations for mitigation measures) was not necessary [91]. The EMA concluded that the overall risk of rVSVΔG-ZEBOV-GP to human health and the environment is negligible, given limited viral shedding, acceptable toxicity in non-human primates, and no horizontal transmission in pigs [84]. Of note, the ERA developed for rVSVΔG-ZEBOV-GP applies specifically to the European countries under EMA purview—other geographic settings, with different native wildlife and domestic animals, low-resource health care systems, and less developed economies (e.g., more heavily dependent on agriculture) may require different starting considerations, supportive datasets, and recommended mitigation strategies.

## 4. Applying the Lessons Learned

### 4.1. Accelerating the Development of Live Recombinant Vaccines Overall

Vaccine developers may not be overly familiar with environmental assessment requirements for GMO medicinal products, nor with their practical application. The experience gained from conducting the ERA for rVSVΔG-ZEBOV-GP may be useful to the clinical development of other live recombinant vaccines, in particular those that must be developed under accelerated timelines in order to combat rapidly emerging infectious diseases (e.g., COVID-19) [41,92,93]. The regulatory requirements for environmental assessments of replication-competent GMO vaccines have to be considered upfront [71,73], along with a plan to proactively generate the data necessary to support ERAs in a timely fashion. Lack of early planning can otherwise lead to delays in starting clinical trials and making a potentially effective vaccine available for use.

From a scientific perspective, in order to support environmental assessments for a live recombinant vaccine, the following need to be generated in parallel with preclinical and clinical development:Robust viral shedding and viremia data, generated from clinical and non-clinical studies using qualified methods that are applied consistently across clinical trials for the candidate vaccine. These data are essential from vaccine recipients and are also important from other species potentially susceptible to infection with the recombinant vaccine.Experiments that evaluate the likelihood for replication in potentially susceptible species and for transmission by biological and/or mechanical vector species.Studies on genetic stability of the vaccine virus and the potential for recombination with other viruses.Robust protocols for vaccine storage, transport/distribution, administration, safe handling, and disinfection/decontamination, all with the goal of preventing (a) contact with the vaccine (other than actual inoculation) and (b) release of the vaccine into the environment.Appropriate risk mitigation strategies.

From a regulatory perspective, we recommend that the following points be considered in preparing the environmental assessment package:Knowledge of each country’s expectations and guidelines for relevant environmental assessments and having discussions with regulatory agencies prior to filing, in order to make sure that expectations will be met.Proper presentation of the environmental assessment with relevant appendices following applicable country guidelines.The likely requirement of detailed information on vaccine packaging and transport.The likely requirement of detailed handling instructions for the vaccine.The environmental assessment should be fully aligned with the proposed product label for the vaccine.

Our experience with preparing the ERA for rVSVΔG-ZEBOV-GP also offers some important lessons for pandemic or public health emergency preparedness, such as the current COVID-19 pandemic. In the case of a large-scale novel disease outbreak or a serious pandemic caused by a known, but previously rarely occurring pathogen, it is imperative to reduce the time necessary to deploy a safe and effective vaccine as much as possible. In case of candidate live recombinant vaccines, these are subject to environmental risk assessment regulations for GMO medicinal products—without environmental assessments that meet regulatory standards, clinical trials cannot be conducted nor marketing authorizations obtained. In order to reduce the time required to generate the supporting data and the environmental assessment itself, vaccine developers may wish to leverage data from similar vaccines (e.g., those using the same vector platforms) or information from the existing literature as much as possible. Further, governments could opt to derogate laws and regulations requiring environmental assessments for GMO vaccines during a major health emergency. For instance, the European Union passed new regulation exempting clinical trial and marketing authorization applications for GMO-based COVID-19 vaccines from the ERA requirement, for as long as the WHO considers COVID-19 to be a pandemic. The new regulation still requires sponsors to minimize predictable environmental impacts resulting from (intended or unintended) release of the recombinant vaccine [94]. Importantly, under this regulation, risk management is the responsibility of each individual member state where a clinical trial takes place or where the vaccine receives marketing authorization [94].

In order to facilitate development and approval of novel GMO vaccines, certain improvements in the regulatory process surrounding environmental assessments would be of great benefit. For instance, harmonized environmental review guidance around the world, streamlined approaches for developing/submitting environmental aspects of clinical trial applications, the ability to refer to other parts of the drug filing in the environmental assessment (to avoid duplication of information), and the option of keeping select information confidential (rather than include in public documents) would greatly simplify and accelerate this important step of vaccine development.

### 4.2. Accelerating the Development of rVSV-Based Vaccines Specifically

Much of the information presented in this article is supportive of the development of future vaccine candidates based on the rVSVΔG vector system platform. Different pathogen-derived envelope (glyco)proteins, when engineered into the VSV vector, may result in recombinant viruses that vary from each other and from wtVSV in their host and cell tropism; there may also be differences in other factors relevant to environmental assessments. Nevertheless, the ERA for rVSVΔG-ZEBOV-GP provides a benchmark for acceptability of similar data from other chimeric constructs utilizing the rVSVΔG platform. For example, researchers might consider testing the replication competence and transmissibility of a new rVSVΔG -based construct in the same insect cell lines that rVSVΔG-ZEBOV-GP was evaluated in, along with conducting biodistribution studies in the same species as was done with rVSVΔG-ZEBOV-GP [68]. The results from those studies could then be compared to the corresponding data for rVSVΔG-ZEBOV-GP. If the virulence and tropism of a new vaccine using the rVSVΔG backbone is similar to that of rVSVΔG-ZEBOV-GP, then levels of shedding and effects on the environment are likely going to be the same as well. In addition, if clinical studies show limited viremia and shedding of such a new rVSVΔG-based vaccine, additional studies to evaluate transmission to other animal species may not be needed.

## 5. Conclusions

rVSVΔG-ZEBOV-GP is a safe, effective, single-dose EVD vaccine with a positive benefit–risk profile that has been shown to rapidly induce high levels of protection in the real-world setting of an actual EVD outbreak. Since rVSVΔG-ZEBOV-GP is a live, recombinant virus, some regulatory agencies required detailed environmental risk assessments prior to initiating clinical trials and also as part of granting marketing authorization. The most in-depth of those assessments was the ERA conducted for the EMA. A number of potential risks to human and environmental health were considered, all under the conditions of the proposed use of the vaccine. The most important consideration was the likelihood of rVSVΔG-ZEBOV-GP becoming persistent and invasive through direct and/or indirect interactions between the vaccine and non-target organisms (i.e., potential animal hosts).

Based on data from a number of preclinical and clinical studies that explored the likelihood of various potential hazards, the environmental risks associated with rVSVΔG-ZEBOV-GP appear to be negligible. The overall risk for non-vaccinated individuals, and for the environment at large, can be considered negligible. Clinical data demonstrate that rVSVΔG-ZEBOV-GP is only infrequently shed from vaccinated individuals and if so, then not at significant levels. In addition, viremia levels are low and transient. It is therefore highly unlikely that a vaccinated individual would infect potentially susceptible wild animal species, domestic livestock, or human beings with recombinant virus. This conclusion is supported by preclinical data obtained in rodents and pigs. Furthermore, the recombinant vaccine virus does not replicate in, nor appear to be transmissible by, several groups of hematophagous insects known to play a role in the life cycle of wtVSV, the vector that the vaccine is based on. Despite the negligible environmental risks, a number of additional precautions are recommended to further mitigate any potential hazards. These recommendations include covering the injection site and any vesicular rash and for vaccinated individuals to avoid contact with children, immunocompromised people, and livestock for 6 weeks. Some of the environmental assessment components described in this article can inform comparable data for potential future vaccine candidates built on the rVSVΔG vector platform, including in-development vaccines for COVID-19 and emerging infectious diseases.

## Figures and Tables

**Figure 1 vaccines-08-00779-f001:**
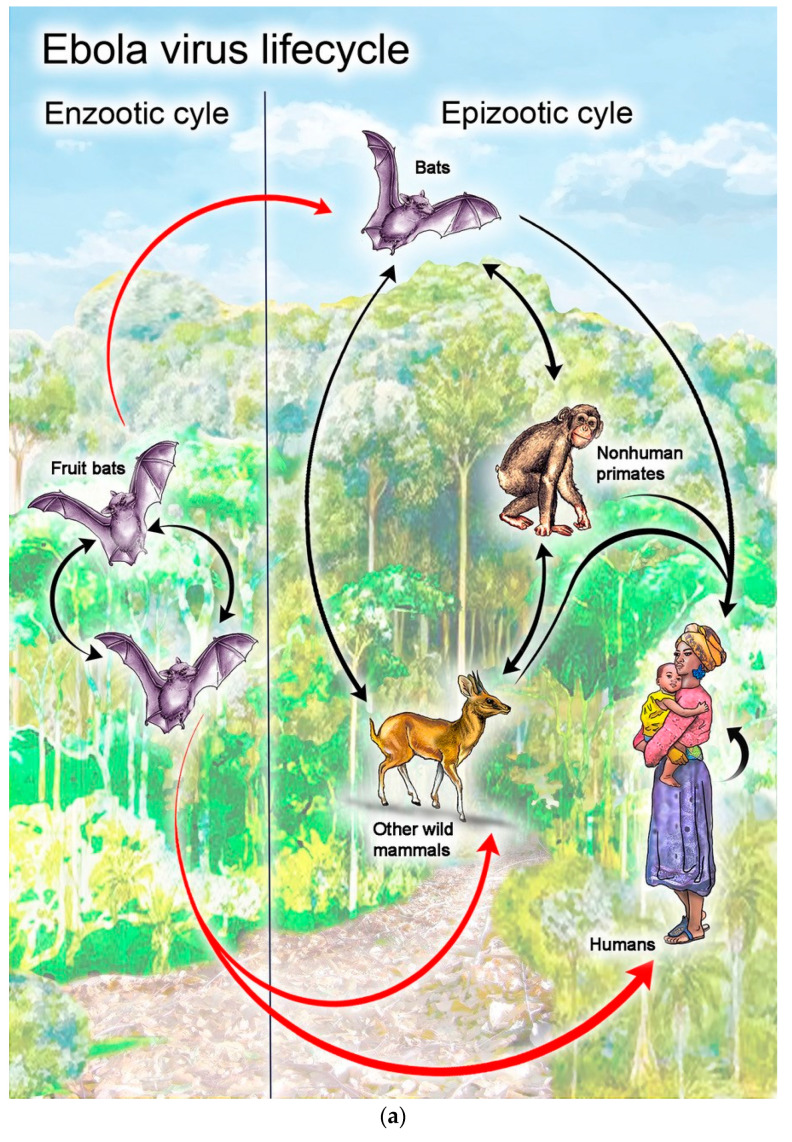
Hypothesized viral life cycle for wild-type (**a**) Ebola virus and (**b**) vesicular stomatitis virus (VSV). Shown are the enzootic cycles (i.e., the predominant, long-term natural host reservoirs of the virus and transmission vectors) and epizootic cycles (i.e., spillover of the virus into other susceptible species and/or alternative transmission vectors). The main natural host reservoir of Ebola virus is thought to be certain species of Old World fruit bats, but the virus is capable of transmission to many other mammalian species [1,7,8,9,10]. The natural host reservoir of VSV is thought to be mainly small rodents, such as deer mice, with sandflies acting as transmission vectors; however, VSV is also capable of infecting many other animal species and can be transmitted via other insects [15,16,17,18,19,20,21,22,23,24,25,26,27,28,29,30,31,32]. ^a^ Pool feeders, i.e., hematophagous arthropods that lacerate their host’s blood vessels and then consume the pooled blood. ^b^ Vessel feeders, i.e., hematophagous arthropods that consume blood by directly inserting their mouthparts into the lumen of their host’s capillary blood vessels.

**Figure 2 vaccines-08-00779-f002:**
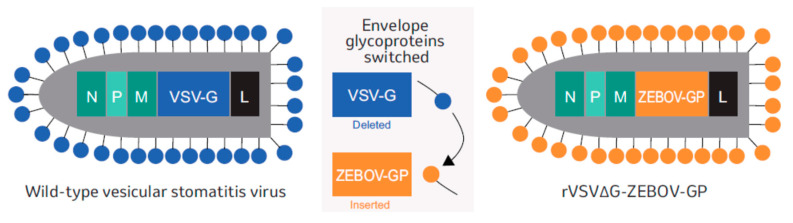
Molecular structure of the chimeric rVSVΔG-ZEBOV-GP live, recombinant vaccine compared with wtVSV. L, large protein. M, matrix protein. N, nucleoprotein. P, phosphoprotein. VSV-G, vesicular stomatitis virus envelope glycoprotein. ZEBOV-GP, Ebola virus envelope glycoprotein.

**Figure 3 vaccines-08-00779-f003:**
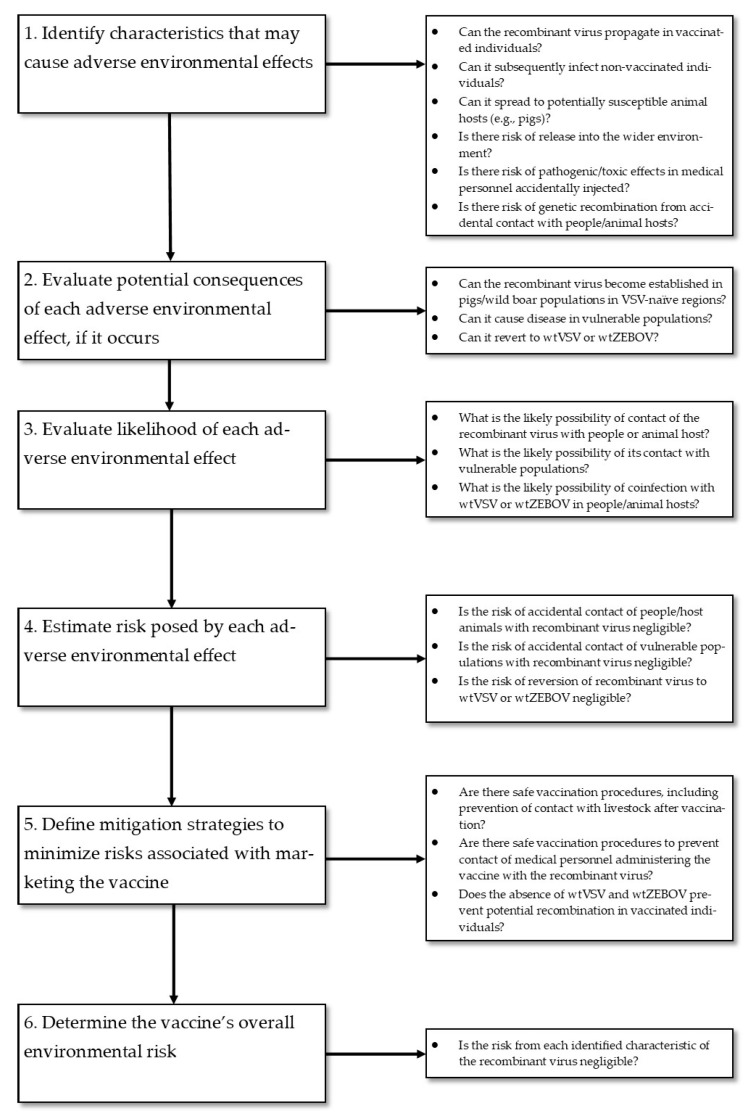
Steps required to conduct an ERA for a GMO medicinal product under EMA regulations (**left**-hand column), illustrated by specific assessment examples from the ERA for rVSVΔG-ZEBOV-GP (**right**-hand column).

**Figure 4 vaccines-08-00779-f004:**
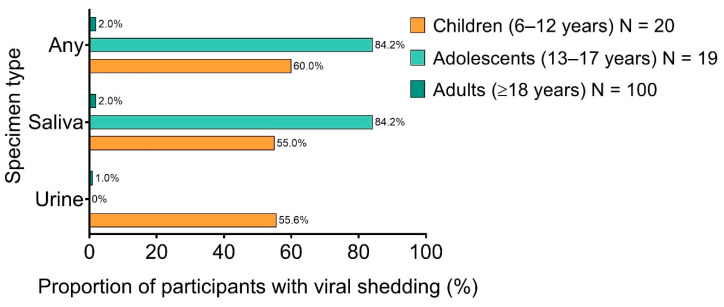
rVSVΔG-ZEBOV-GP shedding and excretion rates among participants in the V920-007 phase 1 clinical trial for whom corresponding data were available.

**Figure 5 vaccines-08-00779-f005:**
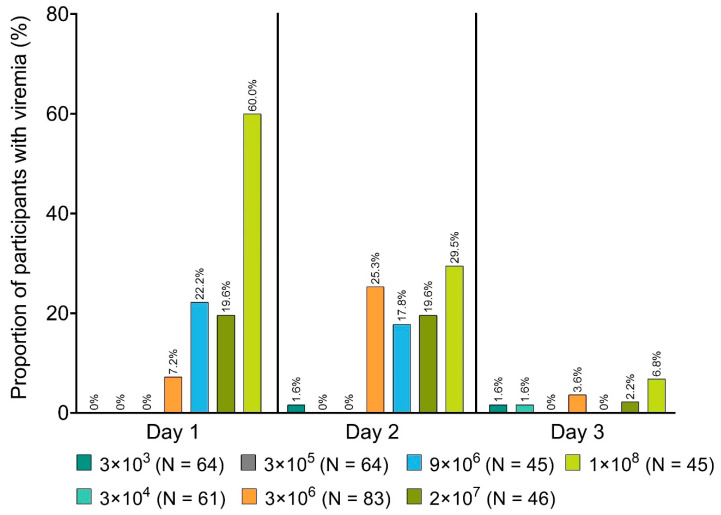
rVSVΔG-ZEBOV-GP viremia rates, over the first 3 days post-inoculation, among participants in the V920-004 phase 1 clinical trial.

**Table 1 vaccines-08-00779-t001:** Comparison of environmental risk assessments required by the EMA for medicinal products that are small molecules and those that are GMOs.

Small-Molecule Medicinal Product	GMO Medicinal Product
Hazardous characteristics Ecotoxicity testing on fish, daphnia, and algae	Hazardous characteristics Pathogenicity, tumorgenicity Replication, invasiveness Horizontal gene transfer
Type and extent of release PEC calculated using standard equations based on dosing, market penetration, and typical wastewater use	Type and extent of release Consideration of viremia and shedding data from non-clinical and clinical trials
Hazard effect levels/concentration PNEC determined based on toxicity testing and safety factors	Hazard effect levels/concentration Comparison to clinically relevant quantities
Potential for persistence and bioaccumulation Abiotic and biotic degradation Octanol–water partition coefficient Fish bioconcentration test	Potential for persistence and bioaccumulation Replication Presence of host species Replication in host species Transmission by non-host organisms Environmental inactivation Climatic conditions
Quantitative assessment based on PEC/PNEC ratios	Qualitative assessment based on likelihood of above factors in natural environment
Labeling Only if risk is identified	Labeling Safe handling “This medicine contains a GMO” statement

ERA, environmental risk assessment. GMO, genetically modified organism. PEC, predicted environmental concentration. PNEC, predicted no-effect concentration.

**Table 2 vaccines-08-00779-t002:** rVSVΔG-ZEBOV-GP clinical trials in which viremia and viral shedding/excretion were evaluated and which were included in the ERA.

Study Name, Location, Literature Citation	Number of Participants	Dose Levels Evaluated, pfu	Viremia Observed	Shedding Observed
***Phase 1***				
V920-001—USA [75]	30	3 × 10^6^, 2 × 10^7^, 1 × 10^8^	Y	Y
Caco-2V920-002—Caco-2USA ^a^ [75]	30	3 × 10^6^, 2 × 10^7^, 1 × 10^8^	Y	Y
Caco-2V920-003—Caco-2Canada [46]	30	1 × 10^5^, 5 × 10^5^, 3 × 10^6^	Y	N
Caco-2V920-004—Caco-2USA [47]	418	3 × 10^3^, 3 × 10^4^, 3 × 10^5^, 3 × 10^6^, 9 × 10^6^, 2 × 10^7^, 1 × 10^8^	Y	Y ^c^
Caco-2V920-005—Caco-2Switzerland [76]	102	3 × 10^5^, 1 × 10^7^, 5 × 10^7^	Y	N
Caco-2V920-006—Caco-2Germany [76]	30	3 × 10^5^, 3 × 10^6^, 2 × 10^7^	Y	N
Caco-2V920-007—Caco-2Gabon ^b^ [77]	155	3 × 10^3^, 3 × 10^4^, 3 × 10^5^, 3 × 10^6^, 2 × 10^7^	Y	Y ^d^
Caco-2V920-008—Caco-2Kenya [76]	40	3 × 10^6^, 1 × 10^7^	Y	N

^a^ In this trial, 2 doses were given 28 days apart. ^b^ This trial also enrolled participants < 18 years of age. ^c^ Only in one participant, who also had the highest viremia level reported in this trial. ^d^ Mostly in children and adolescents.

**Table 3 vaccines-08-00779-t003:** Risk reduction measures recommended in the ERA to further reduce the potential human and environmental hazards associated with rVSVΔG-ZEBOV-GP.

Potential Exposure	Measure
Accidental breakage/spillage during transport or administration	Medical personnel involved in the administration of rVSVΔG-ZEBOV-GP should be wearing personal protective equipment in order to minimize exposure. Disinfectants such as aldehydes, alcohols, and detergents should always be available in case breakage/spillage were to occur, in order to inactivate the vaccine through chemical disinfection and prevent release into the environment. Detailed instructions on how to handle accidental breakage/spillage have been developed and accompany each shipment of rVSVΔG-ZEBOV-GP. Any unused vaccine or waste material should be disposed of in compliance with applicable institutional guidelines for GMOs or biohazardous waste, as appropriate.
Direct human contact with rVSVΔG-ZEBOV-GP shed by vaccinated individuals	Vaccinated individuals should be informed about the potential for shedding and the need to avoid close association with high-risk individuals (i.e., immunocompromised and children < 1 year old) for up to 6 weeks following vaccination, particularly exposing them to blood and bodily fluids. People who develop vesicular rash after receiving the vaccine should cover the vesicles until they heal. The vaccination site or any vesicles should be covered with an adequate bandage (e.g., adhesive bandage, gauze, and tape) that provides a physical barrier protecting against direct contact. The covering may be removed when there is no visible fluid leakage. Vaccinated individuals should not donate blood for 6 weeks following vaccination.
Accidental needle stick injury	The injection site should be disinfected immediately and covered, in the same fashion as advised for vaccinated individuals (see above). If this were to occur in the context of a clinical trial, the injured individual should be followed up for safety in the same fashion as a purposely vaccinated trial participant.
Direct contact of animals with rVSVΔG-ZEBOV-GP shed by vaccinated individuals	Vaccinated individuals should avoid exposing livestock to their blood and bodily fluids for at least 6 weeks following vaccination. The vaccination site or any vesicles should be covered (see above).
Unintended use or misuse	rVSVΔG-ZEBOV-GP shipments delivered to hospital centers for vaccination should be well controlled and be handled as GMOs per local regulations. Only appropriately trained medical personnel should have access to the vaccine. If eye contact were to occur, eyes should be flushed with tepid tap water for 5 min. If skin contact were to occur, exposed areas should be washed with ordinary soap and tap water.

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
