# Peer review of "Environmental Risk Assessment for rVSVΔG-ZEBOV-GP, a Genetically Modified Live Vaccine for Ebola Virus Disease"

_vaccines, 2020, doi:10.3390/vaccines8040779_

Round 1

Reviewer 1 Report

The review by Tell et al., provides a comprehensive analysis of the Environmental Risk Assessment for the rVSV(G)-ZEBOV GP GMO vaccine as required by the EMA.The review is well written and analyzes the ERA thoroughly.

Few comments are below

  1. While quoting the lines from the ERA can be considered acceptable, extensive quoting can be replaced by a conclusion in the authors' own words along with the citation especially in lines 111-112 and lines 356-359. 
  2. Cannot distinguish the dark green and dark blue in figures 4 (adults and children groups) and 5 (3x10^3 and 3x10^6 groups)
  3. LDL is abbreviated but not explained prior to abbreviation.

Reviewer 2 Report

This is a very thorough, extremely well-written paper and will be a useful resource in the field.  I found only a few minor issues:

1) Abstract.  Sentence beginning with "This ERA" should be edited for clarity. 

2) Abstract.  Sentence beginning with "Clinical data"- consider deleting the word "from".

3) Abstract. Sentence beginning with "Animal data"- consider using the word "suggests" instead of "suggested" because mixed tenses are used in that sentence.

4) Line 127- consider deleting the word "only".

5) Line 245- the sentence ending with the word "likelihood"- this seems like an unfinished sentence.

6) Figure on page 12.  Could n= be put into this graph? (or a range of n=).  Also, did dose impact the % of viral shedding?  (An addition with this information could be described/reflected in the graph, legend or text of the manuscript).

7) Figure on page 13.  Could the legend be one line horizontally?  I personally had a black and white print out so this would have made it easier to read.  

8) Figure on page 13.  n= for this graph?  

9) Line 343- Levels of what?  (in reference to the words "at levels not exceeding")

10) Line 372- "Rodents are also natural hosts of both wtEBOV...."  -this needs to be changed because WT rodents are not susceptible to wtEBOV.  The virus needs to be adapted by serial passage in order to productively infected mice or guinea pigs.

Reviewer 3 Report

Tell et al. Describe the Environmental Risk Assesment for the rVSVΔG-ZEBOV-GP vaccine vector, which has been developed for approval of the vaccine by the EMA.

  • Line 36: “deficits” typo?
  • Table 2: The contains the information that shedding and viremia was analyzed in the different clinical trials. However, it does not say if shedding viremia was actually observed or not. This information would be helpful as a summary, especially as in the following paragraphs only the data for selected trials (V920-007 + V920-004) is discussed.
  • Line 369 + 380 + 384: “data on file” What does this mean? Is there some supplementary data for this manuscript?
  • Line 430 ff.: Is there a reference for the genetic stability studies? If not some more details would be nice here, e.g. how many passages, where bulk cultures sequenced or individual plaques etc.
  • Line 432/433: “suggesting a low probability of point mutations occurring.” How has this been determined, see also previous comment. For VSV vectors one would expect a rather high likelihood of point mutations as VSV’s polymerase has no proof-reading function and therefore a rather high likelihood of introducing mutations. Of course not all of these mutations will result in amino acid exchanges and as the ZEBOV GP is essential for virus replication there is a low probability of significant changes in the protein sequence. However, it seems rather unlikely that individual virus clones have no mutations after 5-10 passages. I agree with the overall conclusion of the authors that genetic stability should not be an issue for the rVSVΔG-ZEBOV-GP vaccine vector, however, the authors could discuss here that this might be different for other VSV-based vaccine vectors where the vaccine antigen is not essential for virus replication (see Quiñones-Kochs et al. 2001, PMID: 11531419).
  • Table 3: It is not completely clear if the recommendations are the conclusion of the authors or requirements by the EMA. The recommendation to avoid close contact to children for 6 weeks after vaccination does not seem to be really practicable as many persons in the target population of the vaccine will have children for which they will need to care. Additionally, the authors could discuss recent studies where the rVSVΔG-ZEBOV-GP vaccine vector has been applied in pregnant women or children.
  • References: Some references do not seem to be formatted correctly, e.g. 8, 21, 35, 89, 90, 93
